# Online Dense Video Captioning with Factorized Action Object Retrieval

## Abstract

Dense video captioning presents the dual challenge of temporally localizing events and generating descriptive captions within long videos. However, existing methods often struggle to handle evolving contexts in streaming settings or depend on static, global retrieval mechanisms. To address these limitations, we introduce a novel framework that embeds a dynamic, factorized retrieval mechanism directly into a causally-aware video processing backbone. Unlike approaches utilizing static global retrieval, our method dynamically retrieves concise action and object phrases at each timestep as the video streams. These retrieved phrases are integrated into a causal, autoregressive transformer, enriching the video representation to enhance the text decoder. Furthermore, to mitigate the scarcity of densely annotated video data, we introduce an image-based simulated video pretraining strategy. Experiments on the ViTT, YouCook2, and ActivityNet benchmarks demonstrate that our model significantly outperforms existing global and online methods.

## 1 Introduction

As video content continues to grow exponentially, the need for automatic and detailed video understanding has become increasingly critical. Dense video captioning (Zhu et al., 2022; Wang et al., 2021a; Yang et al., 2023) is the task of generating multiple, temporally localized descriptions for events in long videos, which is crucial for applications like video search, summarization, and accessibility. However, many traditional methods generate a single caption for an entire video. This may not effectively capture the temporal granularity of individual events as they occur

Dense video captioning is essential for tasks like video search and summarization, requiring models to generate localized descriptions for events in long videos. However, most existing methods (Zhu et al., 2022; Wang et al., 2021a; Yang et al., 2023) operate offline, requiring access to the entire video file. This global paradigm is incompatible with streaming settings where data arrives sequentially and future context is unavailable. For such applications, the model must process an continuously evolving context without access to future frames.

Recent works (Zhou et al., 2024; Piergiovanni et al., 2024) have introduced online dense video captioning to address these streaming constraints. Yet, these models often struggle to capture precise semantics relying solely on visual features. While Retrieval-Augmented Generation (RAG) has proven effective for offline video understanding (Xu et al., 2024; Kim et al., 2024), standard RAG methods rely on global retrieval over the full video duration. This breaks the causal requirement of streaming. To address this, we introduce a stream-aligned framework that integrates retrieval directly into the online video processing.

To address these challenges, we introduce a novel Online Action-Augmented Dense Video Captioning framework that fundamentally intertwines retrieval with the causal progression of the video. Our core contribution is a dynamic, segment-level retrieval mechanism that is causally integrated with the video representation. As the video streams, our model processes it in an incremental manner, performing two key actions at each timestep: 1) Factorized retrieval: Instead of retrieving lengthy captions, our model retrieves concise and factorized action and object phrases from two distinct corpora. This decoupled design allows for more flexible and compositional pairings to describe a wider array of events. 2) Causal integration: These retrieved

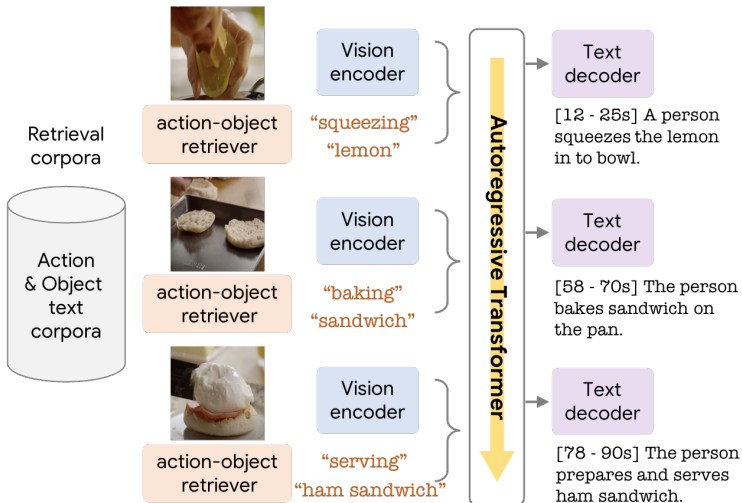

Figure 1: Our model tackles online dense video captioning by dynamically retrieving action-object phrases from a preconstructed corpus as it incrementally processes video segments. This approach dynamically integrates visual and textual cues to produce accurate, temporally aligned captions that capture the evolving actions within the video.

phrases are immediately fused with the visual features and processed by an autoregressive transformer. This ensures that the model's understanding at any given moment is grounded in a retrieval-augmented history, enabling it to generate more accurate and contextually-aware captions for the current segment. Our approach provides timely, relevant context while avoiding the limitations of offline, global retrieval.

Furthermore, we address the scarcity of large-scale dense video caption datasets by exploring image-based simulated video pretraining. We leverage image-text paired data to align pretraining with online video captioning and improve model performance. Experiments on the ViTT, YouCook2, and ActivityNet benchmarks show significant performance gains, highlighting the effectiveness of our approach. We will make our code publicly available upon acceptance.

## 2  Related work

**Dense video captioning.**  The goal of dense video captioning is to provide detailed, temporally-aligned descriptions of multiple events within a video. Unlike conventional methods that produce a single caption for an entire video (Wu & Krahenbuhl, 2021; Sun et al., 2022; Ashutosh et al., 2023; Gao et al., 2023; Cheng & Bertasius, 2022; Islam & Bertasius, 2022; Lin et al., 2022; Zhang et al., 2019), dense captioning is particularly beneficial for understanding long, untrimmed videos. Early approaches often employed a two-stage method, detecting event boundaries before generating captions (Iashin & Rahtu, 2020). More recent work has shifted towards unified, end-to-end models that jointly predict timestamps and captions (Wang et al., 2018; Zhang et al., 2022; Zala et al., 2023; Yang et al., 2023; Liu et al., 2025; Wu et al., 2025). While large multimodal video LLMs (Lin et al., 2023; Song et al., 2024; Zhang et al., 2023; Li et al., 2023; Ren et al., 2024) have emerged, they typically underperform dedicated state-of-the-art dense captioning methods.

**Online dense video captioning.**  Crucially, most existing models operate *offline*, requiring access to the entire video for processing. In contrast, *online* video understanding focuses on predicting actions and timing without access to future frames (*e.g.* online action detection (De Geest et al., 2016; Wang et al., 2021b; Kondratyuk et al., 2021; Zhao & Krähenbühl, 2022; Zhao et al., 2023), temporal action localization (Singh et al., 2017; Buch et al., 2017; Kang et al., 2021), and video dialogue (Chen et al., 2024)). Zhou et al. (2024) recently pioneered online dense video captioning by continually clustering tokens from the video stream. Our work diverges from these token-aggregation strategies; instead, we introduce online retrieval augmentation to an autoregressive model, incorporating timely external knowledge about actions and objects to enrich

the model's contextual understanding. Furthermore, distinct from recurrent masked autoencoders Zoran et al. (2026) which rely on RNN-style architectures for pixel-level reconstruction, our method employs an autoregressive Transformer for next-token text prediction. Our contribution lies not just in processing video sequentially, but in the dynamic, timestep-level retrieval and integration of factorized text priors directly into this streaming captioning.

**Retrieval-augmented methods.** Augmenting models with external knowledge through retrieval has become a popular technique in vision-language tasks, like video retrieval (Zhang et al., 2021; Jing et al., 2023; Chen et al., 2023), pretraining (Xu et al., 2021). In video captioning, retrieval-augmented methods (Xu et al., 2024; Kim et al., 2024) often improve generation by fetching full-sentence captions to serve as a reference. This retrieval process is typically done at a global level for the entire video, or by combining segment-level retrievals into a single, static context. In contrast, our method performs online, factorized retrieval of concise action-object phrases. This allows for a more dynamic and flexible integration of contextual priors at each timestep, which is better suited for online dense video captioning. Furthermore, recent works have explored decoupled modeling for video understanding, such as learning local semantic signals for foreground-background separation in the visual space for video anomaly detection Wang et al. (2026). In contrast, our approach operates via decoupled factorized retrieval specifically within the textual space (separating actions and objects). While prior action retrieval methods (*e.g.*, Action-semantic consistent knowledge Wang et al. (2024b), Semantic query learning Wang et al. (2024c), and Event completeness learning Wang & Chen (2025)) focus on offline or global approaches for temporal action localization, our framework introduces a dynamic, online retrieval mechanism at the frame/segment-level.

## 3 Method

### 3.1 Preliminaries

#### 3.1.1 Captioning model

Our model architecture is built upon the CLIP (Radford et al., 2021), a foundation model also leveraged by other video captioning methods (Yang et al., 2023; Zhou et al., 2024). A single, frozen CLIP ViT serves as our shared vision encoder. Its features are used for both the retrieval and captioning pathways. For the text-side, we utilize two separate instances of the CLIP text model. 1) text encoder for retrieval: One copy of the CLIP text model is kept frozen and functions as a standard text encoder. Its only purpose is the one-time, offline pre-computation of text embeddings for our action-object retrieval corpus, as will be detailed in section 3.3. 2) Text decoder for captioning: A second copy of the CLIP text model is adapted into our text decoder. This model is not frozen. We first modify it with causal attention masking to enable autoregressive text generation. Then, we further train it on the LAION-2B dataset for the image captioning task. This process effectively transforms it from a contrastive encoder into a generative decoder, which is then trained for the final dense video captioning task.

#### 3.1.2 Dense video captioning

Given a video $V \in \mathbb{R}^{T \times H \times W \times 3}$, our goal is to generate a set of temporally localized captions: {( [$s_1$][$e_1$][caption text$_1$] ), ..., ( [$s_n$] [$e_n$] [caption text$_n$] )}, where start [s] and end [e] times mark the event boundaries of each caption. Inspired by Vid2Seq (Yang et al., 2023), we represent the start [s] and end [e] times as discrete vocabulary tokens directly within the text sequence. This unified format allows the model to generate both the temporal boundaries and the descriptive caption in a single output stream.

### 3.2 Online Captioning with a Causal Video Model

#### 3.2.1 Autoregressive video processing.

Unlike global offline methods which are computationally expensive for long videos, our autoregressive model processes video frame-by-frame in an online fashion. As illustrated in Figure 2, each frame passes through a frozen CLIP ViT, which extracts $M$ tokens. A Token Aggregator then condenses these to $N$ tokens ($N \ll M$).

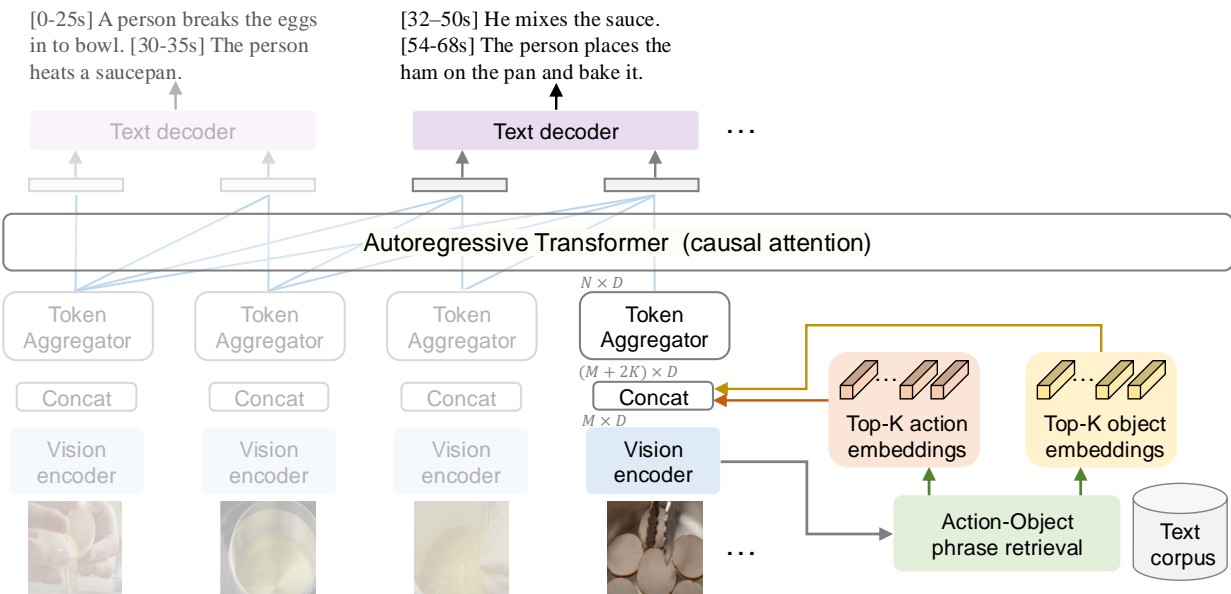

Figure 2: **Overview of our online action-object augmented dense video captioning.** Our model processes video frames incrementally, retrieving top-K relevant action-object phrases from a pre-constructed text corpus. These retrieved phrases are integrated autoregressively into the video representation. Caption generation occurs at the segment level (every multiple frames), where each segment's frame features are causally contextualized before decoding, enhancing coherence across multiple segments.

As new frames arrive, their features are stacked and processed by an Autoregressive Transformer, which applies causal attention along the temporal axis. This incrementally enriches each frame's representation with causally-aware historical context.

### 3.2.2 Segment-based text decoding.

For text generation, we employ segment-based decoding instead of global decoding, generating captions every $L$ frames (a 'segment'). This results in $S = T/L$ decoding steps for a video with $T$ frames. As shown in our experiments Table 2, this segment-based approach outperforms global decoding because it simplifies the generation task at each step while maintaining long-term context through the causally-aware video features. It also helps preventing the over-summarization often seen in global methods.

An event is assigned to a segment if its end time [e] falls within that segment's interval, which allows the model to generate captions for events that span multiple segments, *e.g.*, start times [s] possibly in earlier segments and end times [e] within the current segment (see Figure 4). Segments containing no events are labeled as "[BOS][EOS]". When multiple actions occur, their captions are concatenated sequentially within the target sequence, *e.g.*, "[BOS][s$_1$][e$_1$][caption text$_1$][s$_2$][e$_2$][caption text$_2$] ... [EOS]". Because the underlying visual features have been causally processed, the model remains aware of prior events, ensuring coherence throughout the video.

### 3.3 Dynamic Retrieval and Integration

We propose to enhance online dense video captioning by dynamically retrieving and integrating relevant action-object priors for each video frame. This method introduces a dynamic online retrieval mechanism combined with autoregressive modeling, where action-object priors are retrieved and incorporated at each timestep as the video streams.

| Pipeline Phase | Dataset | Modality Used | Role |
|---|---|---|---|
| **Pretraining** | **LAION-2B** | Image + Text | ViT initialization /Decoder pretraining & Simulated Video Pretraining |
| **Dictionary Construction** | **Kinetics-700, UCF-101, EpicKitchen, SSV2** | Text-only (Labels) | Offline Action Phrase Corpus Construction |
| | **V3Det, LVIS, Places365** | Text-only (Labels) | Offline Object Phrase Corpus Construction |
| | **HowTo100M** (Appendix) | Text-only (Subtitles) | Alternative Action Corpus Construction |
| **Finetuning & Eval** | **ViTT, YouCook2, ActivityNet** | Video + Text | Dense Video Captioning |

Table 1: **Dataset usage and pipeline.** We separate the datasets used across our pipeline into distinct phases of pretraining, dictionary construction, and evaluation.

### 3.3.1 Construction of action and object phrase corpus.

To enable retrieval, we first construct a corpus of concise action and object phrases. We propose a factorized retrieval approach that independently retrieves action and object phrases from two distinct corpora, rather than relying on lengthy raw video captions like (Xu et al., 2024). We construct our static, decoupled text corpora using independent video and image datasets that are not mixed with our dense video captioning evaluation benchmarks. Table 1 outlines the specific modality and role of every dataset utilized in our pipeline. The action and object phrases are collected from multiple recognition datasets by using their labels (text) information only. Notably, some action categories inherently include objects (*e.g.* separating egg), while others do not (*e.g.* clapping). In contrast, the object phrase corpus consists of distinct object-focused categories, independent of specific actions.

This decoupled design allows for flexible and diverse action-object pairings. It uses a modest corpus of under 20k text embeddings, which mitigates the burden on storage and memory. Furthermore, the retrieval process is highly efficient, taking less than $1ms$ per query with FAISS Johnson et al. (2019) and adding negligible overhead.

### 3.3.2 Precomputation of text embeddings.

To optimize retrieval efficiency, we precompute text embeddings for all action and object phrases using a frozen CLIP text encoder. This precomputation is performed only once, ensuring efficient retrieval without redundant computations.

### 3.3.3 Online retrieval and integration of action-object priors.

Our vision encoder processes the video frame-by-frame. Frame features are globally pooled and compared to the precomputed text embeddings using cosine similarity, to retrieve top-K action and object phrases. These retrieved action and object embeddings are then concatenated with the original visual features, forming an enriched representation of shape $(M + 2K) \times D$. This fused multimodal features are passed through the Token Aggregator and then Autoregressive Transformer, resulting in a causally-aware representation that is critical for dense video captioning task.

### 3.3.4 Mixed training strategy.

To improve generalization and robustness, we employ a mixed training strategy. During training, we replace the retrieved text embeddings with a learnable [none] embedding 50% of the time. This ensures the model can generate captions effectively both with and without retrieval augmentation.

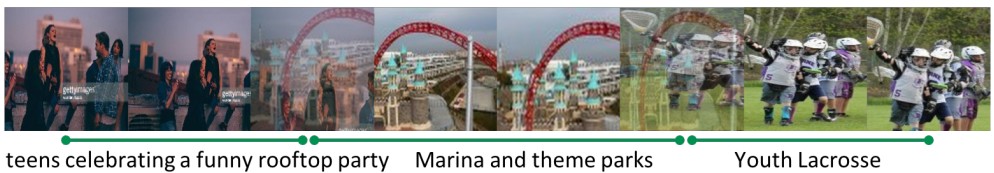

teens celebrating a funny rooftop party     Marina and theme parks     Youth Lacrosse

Figure 3: Simulated video pretraining stitches 3-5 images (repeated/blended) into 16-frame sequences. We visualize 8 frames example for brevity.

### 3.3.5 Frame construction strategy.

To better capture visual content at each timestep, we explore a frame construction strategy that tiles multiple frames into a spatial grid. In a standard approach, frames would be processed independently in sequence $f_i$, $f_{i+1}$, $f_{i+2}$, ...

In contrast, our strategy uses a sliding window of size 4 with a stride of 1 to create composite images. Specifically, four consecutive 256×256 frames are spatially tiled into a single 2×2 grid to form a 512×512 image. The sequence of inputs to the vision encoder thus becomes like {Composite image of ($f_i$, $f_{i+1}$, $f_{i+2}$, $f_{i+3}$)}, {Composite image of ($f_{i+1}$, $f_{i+2}$, $f_{i+3}$, $f_{i+4}$)}, {Composite image of ($f_{i+2}$, $f_{i+3}$, $f_{i+4}$, $f_{i+5}$)}, ... and so on. If fewer than four frames are available (*e.g.*, at the start of the video), we pad by repeating the first frame. The extracted features from these composite images are then used for both retrieval and caption generation.

This approach better aligns with our CLIP encoder's pretraining on static images. While our model is not trained on video-text datasets, presenting temporally related frames within a single input may help the frozen encoder capture inter-frame relationships more effectively than processing frames independently.

As shown in Table 10, this strategy provides a consistent performance boost on all benchmarks when added to our model already pretrained with simulated video (Ours b.).

### 3.4 Simulated Video Pretraining

To address the limited availability of densely captioned video datasets, we explore an image-based pretraining method. Unlike Vid2Seq (Yang et al., 2023), which uses large-scale video data with ASR-generated pseudo captions, our approach uses the LAION-2B image-text dataset to simulate video sequences. We stitch together 3 to 5 images, repeating each multiple times to form a 16-frame sequence. To create smoother transitions and add variability, we blend the pixels at image boundaries. Furthermore, we apply `torchvision.transforms.RandomResizedCrop` with a scale of (0.8, 1.0). To mimic camera motion, we implement a temporally-aware crop that interpolates crop parameters between the first and last frames, creating a continuous motion effect (see Figure 3). While semantic coherence is not explicitly enforced between stitched images, this approach effectively teaches temporal localization and captioning from static images. This allows us to bypass the need for massive (100M-1B scale) video-caption pretraining datasets commonly used by other methods (Yang et al., 2023; Zhou et al., 2024; Wu et al., 2024).

The primary goal of this simulated pretraining is not to enforce semantic coherence across the stitched scenes. Rather, it is designed specifically to teach the model's temporal localization mechanism (*i.e.* generating accurate start [s] and end [e] tokens) and to train the autoregressive attention to handle changing visual contexts. By stitching independent, random images (as illustrated in figure 3), we simulate sharp event boundaries. This forces the model to recognize when one visual event terminates and a completely new one begins, serving as a highly effective temporal grounding proxy before training on actual video data.

Each synthetic sequence is paired with its corresponding image captions, with the temporal span of each image defined by the stitching process, replicating the dense video captioning format: $\{(\ [s_1][e_1][caption\ text_1]\ ),\ ...,\ (\ [s_n]\ [e_n]\ [caption\ text_n]\ )\}$. This pretraining provides a strong warm start for the newly added Autoregressive Transformer and Token Aggregator, using only widely available image-level data.

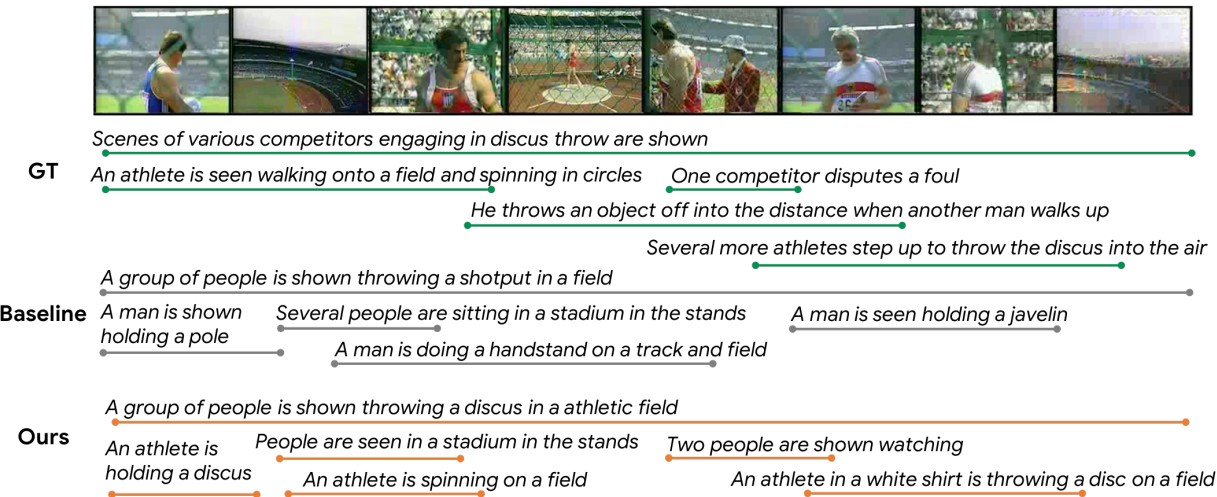

Figure 4: Dense video captioning results of our method. Ground truth captions and their timestamps (green), non-augmented baseline (gray) and our action-augmented model prediction (orange). Our model generates more accurate and temporally aligned captions, *e.g.* 'throwing discus' is well localized in time and integrated in caption.

## 4 Experimental Results

**Datasets and Metrics.** We evaluate on three widely-used dense video captioning benchmarks: ViTT (Huang et al., 2020), YouCook2 (Zhou et al., 2018a) and ActivityNet Captions (Heilbron et al., 2015). We use standard metrics: SODA (Fujita et al., 2020) for overall performance assessing both temporal alignment and caption accuracy, CIDEr (Vedantam et al., 2015) (averaged over IoU thresholds from 0.3 to 0.9), METEOR (Banerjee & Lavie, 2005), and F1 score.

**Implementation Details.** Our model contains approximately 500M parameters and is built from CLIP components pretrained on the LAION-2B dataset (Schuhmann et al., 2021). The main architecture consists of a frozen 303M ViT-Large vision encoder and a 128M text decoder. As detailed in section 3.1.2,, this text decoder is a distinct copy of the CLIP text model, which we adapt for generation by pretraining it for 0.2 epochs on the LAION-2B image captioning task. A separate, frozen copy of the CLIP text encoder is used only once for the offline precomputation of our retrieval corpus.

Our additional components for dense video captioning are lightweight. The Token Aggregator (4 layers, 16M parameters) applies attention pooling at the end to reduce visual features to $N = 32$ tokens per frame. The Autoregressive Transformer (8 layers, 32M parameters) processes video frames incrementally using causal attention.

For our image-based pretraining, we sample 3 to 5 images from LAION-2B, repeating each image multiple times to form 16-frame sequences. The entire video captioning model is trained for 100,000 steps. For the final dense video captioning training, we sample $T = 64$ frames per video at 256×256 resolution. We apply segment-based decoding every 4 frames (16 decoding steps total). The model is finetuned for 20,000 steps with a batch size of 8, taking about 12 hours on 16 devices. The ViT is kept frozen throughout the pretraining and finetuning. At inference, we follow prior works Yang et al. (2023); Zhou et al. (2024) and use beam search with a beam size of 4 to generate the top-1 caption for each decoding step.

### 4.1 Establishing a Baseline Model

We first present our baseline model for online dense video captioning and compare it with a global, offline counterpart. Although our approach shares the principle of online processing with the recent StreamingDVC method (Zhou et al., 2024), we independently establish our own baseline as their model is not publicly avail-

| method | video encoding | text decoding | S | C | M |
|---|---|---|---|---|---|
| Offline | global attention | global | 9.2 | 23.5 | 5.6 |
| Offline | causal attention | global | 8.4 | 21.7 | 5.2 |
| Offline | global attention | segment-based | 9.7 | 26.3 | 6.1 |
| **Online (ours)** | **causal attention** | **segment-based** | 9.0 | 24.5 | 5.6 |

| # frames | # segments | S | C | M |
|---|---|---|---|---|
| 16 | 16 | 7.8 | 21.8 | 5.0 |
| 64 | 64 | 8.7 | 23.6 | 5.3 |
| **64** | **16** | **9.0** | **24.5** | **5.6** |
| 64 | 8 | 8.9 | 24.1 | 5.5 |

Table 2: **Online vs global captioning baselines.**      Table 3: **Number of frames and segments.**

| method | S | C | M |
|---|---|---|---|
| No retrieval | 9.0 | 24.5 | 5.6 |
| Raw video captions (ViTT+YouCook2+ActivityNet) | 9.6 | 26.8 | 6.2 |
| Action names | 10.0 | 28.4 | 6.7 |
| Object names | 9.8 | 29.1 | 6.6 |
| Union of action & object names | 10.2 | 29.4 | 6.9 |
| Decoupled action & object names | **10.6** | **30.1** | **7.2** |
| Oracle caption phrases | 12.1 | 36.2 | 8.3 |

Table 4: **Retrieval corpus comparison.**

able. Both methods utilize a CLIP ViT-Large vision encoder, but our infrastructures differ. StreamingDVC uses a larger 256M parameter T5-Base (Raffel et al., 2020) text decoder pretrained on Web corpora, whereas we employ a smaller 128M CLIP text model further trained on LAION-2B for image captioning, which proves highly effective. We also omit complex features from their design, like a recursive feedback loop, which did not yield gains in our setup.

**Video encoding and text decoding strategy.** In Table 2, we perform an exploratory comparison between our online model and its global captioning counterpart on the ViTT benchmark. We specifically analyze the effect of video encoding method (global bidirectional vs. online causal attention) and the text decoding strategy (global vs. segment-based). **Global model:** First, our global model serves as a strong offline baseline. It combines global bidirectional attention for video encoding with a global text decoder. This setup allows all frames to attend to each other, assuming access to the entire video. The decoder processes the full video representation in one step to generate a single long caption concatenating all event descriptions and timestamps. **Causal video processing:** Next, we vary the video encoding method to use causal attention while retaining the global text decoder. This setup still generates a single long paragraph for the entire video. The transition from bidirectional to causal attention results in a performance drop, highlighting the inherent challenge of online processing without access to future context. **Factorized segment-based decoding:** Finally, our full online model maintains the causal video encoder but changes the text decoding strategy to be segment-based, as described in section 3.2. Here, instead of generating one length caption for the entire video, the model decodes captions segment by segment. This approach reduces the decoding burden at each step and improves captioning performance, leading to more temporally precise output by aligning the decoder's task with the incremental nature of the encoder.

Our baseline models achieve performance comparable to strong methods like Vid2Seq (Yang et al., 2023) and StreamingDVC (Zhou et al., 2024) (see Table 10). This provides a reasonable foundation for evaluating our main contributions. This ensures that the significant improvements detailed in the following sections stem from our proposed online retrieval augmentation, rather than from an inherently superior baseline architecture.

## 4.2 Ablation Studies

We ablate key components of our method on the ViTT dataset in Tables 2-9.

| method | S | C | M |
|---|---|---|---|
| Our online retrieval | **10.6** | **30.1** | **7.2** |
| 25% random phrases | 10.4 | 29.8 | 7.1 |
| 50% random phrases | 10.1 | 29.4 | 6.9 |
| 100% random phrases | 8.6 | 23.9 | 5.4 |

Table 5: **Robustness to retrieval noise.**

| method | S | C | M |
|---|---|---|---|
| Online retrieval | **10.6** | **30.1** | **7.2** |
| Global retrieval | 9.4 | 27.4 | 6.3 |

Table 6: **Online vs global retrieval.**

| top-K | S | C | M |
|---|---|---|---|
| 4 | 9.7 | 28.8 | 6.6 |
| 8 | 10.2 | 29.5 | 6.9 |
| **16** | **10.6** | **30.1** | **7.2** |
| 64 | 10.4 | 30.0 | 7.1 |

Table 7: **Effect of top-K retrievals.**

| method | S | C | M |
|---|---|---|---|
| Gating-based fusion | 10.3 | 29.5 | 6.8 |
| Cross-attention-based fusion | 10.4 | 29.8 | 6.9 |
| **Concatenation** | **10.6** | **30.3** | **7.1** |

Table 8: **Fusion strategies** for integrating retrieved action and object embeddings with visual features

| method | Inference with retrieval | | | Inference without retrieval | | |
|---|---|---|---|---|---|---|
| | S | C | M | S | C | M |
| Non-augmented training | N/A | N/A | N/A | 9.0 | 24.5 | 5.6 |
| Action-augmented training | **10.6** | 30.1 | **7.2** | 4.8 | 17.8 | 3.7 |
| Mixed training | **10.6** | **30.3** | 7.1 | **9.2** | **24.7** | **5.7** |

Table 9: **Mixed training** enhances the model's adaptability, boosting performance in both retrieval-augmented and non-augmented settings.

### 4.2.1 Number of frames and segments.

Table 3 studies the effect of varying the number of frames and segments per video. Overall, using more frames improves performance. The model remains robust across different segment configurations, with 64 frames and 16 segments (*i.e.*, decoding every 4 frames) yielding the best results.

### 4.2.2 Retrieval text corpus.

Table 4 analyzes the effect of different retrieval corpora on model performance. We begin with a "no retrieval" baseline, which performs no augmentation. We then evaluate several retrieval sources: using raw video captions from in-domain datasets (*i.e.*, union of ViTT, YouCook2, ActivityNet training captions), or using corpora containing only action names or only object names (collected as described in section 3.3). To directly test our factorized design, we compare two strategies. The first is a "union" corpus, which combines all action and object names into a single retrieval pool, representing the counterpart to our proposed "decoupled" method. In our method, we retrieve from separate action and object corpora and fuse the results. Our decoupled approach achieves the best performance, and outperforms retrieval using raw, full-sentence video captions from the in-domain datasets (ViTT, YouCook2, and ActivityNet). This demonstrates the effectiveness of our factorized strategy, which allows the model to capture a wider and more flexible range of action-object combinations than retrieving from a single, unified source.

Finally, to estimate a practical upper bound, we conduct an oracle test. For this, we use an LLM (Gemma Team et al. (2024)) to extract key action and object phrases directly from the ground-truth captions of the test videos. This represents an ideal retrieval setting where the corpus is perfectly aligned with the dataset and retrieval accuracy is optimal.

### 4.2.3 Retrieval quality.

To validate the retrieval module independently of caption generation, we evaluate it against oracle ground-truth phrases. The module achieved a Recall@5 of 90.3% and Recall@1 of 78.9%, showing its high effectiveness in retrieving relevant priors.

| method | backbone | ViTT | | | | YouCook2 | | | | ActivityNet | | | |
|---|---|---|---|---|---|---|---|---|---|---|---|---|---|
| | | S | C | M | F1 | S | C | M | F1 | S | C | M | F1 |
| *– VideoLLM-based:* | | | | | | | | | | | | | |
| TimeChat (Ren et al., 2024) | 7B MLLM | - | - | - | - | 3.4 | 11.0 | - | 19.5 | - | - | - | - |
| VTimeLLM (Huang et al., 2024) | 13B MLLM | - | - | - | - | 3.4 | 10.7 | 3.5 | - | 5.9 | 27.2 | 6.7 | - |
| *– Non-LLM-based (<1B params):* | | | | | | | | | | | | | |
| E2ESG (Zhu et al., 2022) | C3D | - | - | - | - | - | 25.0 | 3.5 | - | - | - | - | - |
| MT (Zhou et al., 2018b) | TSN | - | - | - | - | - | 6.1 | 3.2 | - | - | 9.3 | 5.0 | - |
| PDVC (Wang et al., 2021a) | TSN | - | - | - | - | 4.9 | 28.9 | 5.7 | - | 6.0 | 29.3 | 7.6 | - |
| GIT (Wang et al., 2022) | GIT | 7.1 | 15.1 | 3.4 | 32.5 | 3.1 | 12.1 | 3.4 | 17.7 | 5.7 | 29.8 | 7.8 | 50.6 |
| OmniViD (Wang et al., 2024a) | VideoSwin | - | - | - | - | - | - | - | - | - | 26.0 | 7.5 | - |
| Vid2Seq † (Yang et al., 2023) | CLIP | 9.8 | 23.0 | 5.0 | 37.7 | 5.7 | 25.3 | 6.4 | 23.5 | 5.9 | 30.2 | 8.5 | 51.8 |
| DoYou (Kim et al., 2024) | CLIP | - | - | - | - | 5.3 | 31.7 | 6.1 | 33.4 | 6.2 | 33.0 | 8.6 | 55.2 |
| DIBS (Wu et al., 2024) | CLIP | - | - | - | - | 6.4 | 44.4 | 7.5 | 31.4 | 5.9 | 31.9 | 8.9 | 55.6 |
| Streaming ⋆ (Zhou et al., 2024) | CLIP | 10.0 | 25.2 | 5.8 | 35.4 | 6.0 | 32.9 | 7.1 | 24.1 | 6.2 | 37.8 | 10.0 | 52.9 |
| DDVC (Liu et al., 2025) | CLIP | - | - | - | - | 6.7 | 38.8 | 6.9 | 33.7 | 6.6 | 35.5 | 8.6 | 56.1 |
| E2DVC (Wu et al., 2025) | CLIP | - | - | - | - | 5.4 | 34.3 | 6.1 | 28.9 | 6.1 | 33.6 | 8.6 | 56.4 |
| CACMI (Jia et al. (2026)) | CLIP | - | - | - | - | 5.6 | 34.8 | 6.2 | 29.3 | 6.4 | 33.8 | 8.7 | 57.1 |
| StayInYourLane (Baek et al. (2026)) | CLIP | - | - | - | - | 7.1 | 39.2 | 6.1 | - | 6.5 | 35.0 | 8.5 | - |
| a. Ours ⋆ | CLIP | 10.6 | 30.3 | 7.1 | 39.2 | 6.9 | 45.6 | 7.9 | 33.8 | 7.1 | 37.6 | 11.1 | 54.8 |
| b. Ours (a + simulated pretrain) ⋆ | CLIP | 11.1 | 32.6 | 7.7 | 40.8 | 7.5 | 46.2 | 8.2 | 34.4 | 7.5 | 38.3 | 11.9 | 55.5 |
| c. Ours (b + tiled frames) ⋆ | CLIP | 11.4 | 33.5 | 7.9 | 41.1 | 7.9 | 46.8 | 8.5 | 34.6 | 8.0 | 38.9 | 12.8 | 55.8 |

Table 10: **Comparison to the state-of-the-art on dense video captioning.** We evaluate on the ViTT, YouCook2, and ActivityNet benchmarks. We report SODA (S), CIDEr (C), and METEOR (M) for caption quality, and F1 score for temporal localization. Our full method uses 64 frames/video with 4 frames/segment. †: version with visual-only inputs. ⋆: Only ours and Streaming Zhou et al. (2024) allow online captioning. We report the mean scores of 3 independent runs for our models, with the standard deviations of {S: ±0.14, C: ±0.41, M: ±0.13}.

| method | online | video-text pretraining | backbone |
|---|---|---|---|
| E2ESG (Zhu et al., 2022) | N | ∅ | C3D |
| PDVC (Wang et al., 2021a) | N | ∅ | TSN |
| OmniViD (Wang et al., 2024a) | N | Kinetics | VideoSwin + Bart |
| TimeChat (Ren et al., 2024) | N | YT-Temporal, ViTT, ActivityNet, etc. | Eva-CLIP-G + Llama-7B |
| Vid2Seq † (Yang et al., 2023) | N | YT-Temporal-1B | CLIP-L + Bert-B |
| DoYou (Kim et al., 2024) | N | ∅ | CLIP-L |
| DIBS (Wu et al., 2024) | N | Howto100M | CLIP-L |
| Streaming (Zhou et al., 2024) | **Y** | YT-Temporal-1B | CLIP-L + Bert-B |
| DDVC (Liu et al., 2025) | N | ∅ | CLIP-L |
| E2DVC (Wu et al., 2025) | N | ∅ | CLIP-L |
| CACMI (Jia et al., 2026) | N | ∅ | CLIP-L |
| Ours (ours) | **Y** | ∅ | CLIP-L |

Table 11: **Comparison of pretraining data and backbones among state-of-the-art methods.** Our approach achieves superior online performance without relying on massive video-text pretraining datasets or advanced backbones.

### 4.2.4 Online vs global retrieval.

We then evaluate the benefits of our online retrieval mechanism compared to a global retrieval setup. In the global setup, phrases are retrieved from all frames ($T \times K$ retrievals) and then temporally pooled into a global retrieval embedding. As shown in Table 6, our online approach, which retrieves and incorporates information at each timestep, clearly outperforms global retrieval (10.6 vs. 9.4 SODA). This suggests that our online retrieval provides more temporally relevant and localized information.

### 4.2.5 Robustness to inaccurate, noisy retrieval.

To study the impact of retrieval errors, we conducted a robustness analysis in Table 5. The performance degrades only marginally even when 50% of the retrieved phrases are replaced with random noise. While performance is significantly impacted when no relevant phrases are retrieved (100% random), the model is robust to substantial noise as long as some relevant context is available. This resilience may be attributed to our model's design, including the Token Aggregator which learns to weigh inputs based on their relevance, and a mixed training strategy that enhances robustness to varied input quality.

### 4.2.6 Number of retrieved phrases.

Next, we evaluate the effect of top-K retrieval in Table 7. Using 16 retrieved phrases results in the best performance.

### 4.2.7 Alternative retrieval fusion strategy.

To validate our factorized retrieval integration, we explore alternative fusion strategies to combine the visual features with the retrieved action/object embeddings before passing them to the causal transformer. We compare our simple concatenation to gating-based fusion (learning a weighted sum of text and visual features) and cross-attention-based fusion (using visual features as queries to attend to retrieved text keys/values). As shown in Table 8, simple concatenation yields the best empirical performance.

### 4.2.8 Mixed training for non-augmented inference.

Our mixed training strategy alternates between retrieval-augmented and non-augmented training to enhance the model's adaptability for inference without retrieval augmentation. Table 9 shows that mixed training significantly improves the non-augmented inference while maintaining strong performance when augmentation is available.

### 4.2.9 Analysis of caption density.

We observe that our model generates denser captions on the ViTT (9.8 per video) compared to both the ground truth (7.1) and the offline baseline (5.5). This likely reflects valid granular details rather than hallucinations, substantiated by our improvements in precision-sensitive metrics like METEOR which would otherwise decrease. To further verify this, we conducted a blind pairwise evaluation using Gemini 2.5 Pro on ActivityNet. The model is given a video and prompted to penalize false positives and hallucinations. Our method is preferred over the online baseline in 72% of cases, confirming the additional descriptions are accurate.

## 4.3 Comparison to State-of-the-art Methods

We compare our method with state-of-the-art global and online models on the ViTT, YouCook2, and ActivityNet Captions benchmarks. As shown in Table 10, our method outperforms both global and online methods across all benchmarks and metrics.

On the ViTT dataset, our online action augmented model (Ours a.) achieves 10.6 SODA, 30.3 CIDEr, 7.1 METEOR, and 39.2 F1, surpassing the previous best method Streaming (Zhou et al., 2024), with gains of +0.6 SODA, +5.1 CIDEr, +1.3 METEOR, +3.8 F1. These results demonstrate the broad effectiveness of our dynamic retrieval and integration strategy.

Our performance is further enhanced by the proposed simulated video pretraining (Ours b.) which achieves gains of +1.1 SODA scores on ViTT, +1.5 on YouCook2, and +1.4 on ActivityNet, over the previous best methods. Prior methods often rely on large-scale video captioning pretraining. For instance, Vid2Seq Yang et al. (2023), Streaming Zhou et al. (2024) and TimeChat Ren et al. (2024) use YT-Temporal-1B Zellers et al. (2022), while DIBS Wu et al. (2024) creates pseudo-labeled and curated dataset from HowTo100M Miech et al. (2019). In contrast, our model achieves strong results without requiring extensive video captioning pretraining.

While additional performance gains could come from incorporating Automatic Speech Recognition (ASR) as used in Yang et al. (2023); Wang et al. (2021a), we intentionally avoid it. ASR often overlaps with ground truth captions and is closely tied to action occurrences, which can potentially inflate performance metrics without accurately reflecting the model's visual understanding.

We note that an additional tiled frames strategy (section 3.3.5) provides a further performance boost across all datasets.

Table 11 compares various strategies used in existing methods, focusing on key aspects such as support for online video captioning, reliance on video-text pretraining, and the backbone models employed.

**Computational cost.** Our model is computationally efficient. The retrieval process takes **less than $1ms$ per query** with FAISS Johnson et al. (2019). In terms of computational cost, for a 64-frame input, our online model requires 6560 GFLOPs, which is more efficient than our offline counterpart (8320 GFLOPs). This is approximately 2.7 times more efficient than the previous state-of-the-art online method StreamingDVC (17900 GFLOPs).

### 4.4 Visualization

Figure 4 presents the results of our method on the ActivityNet dataset. Compared to the baseline, our online model produces more temporally aligned and accurate captions, such as identifying actions like 'throwing discus'. This shows the effectiveness of online action retrieval and integration.

## 5 Broader Impact

As discussed in section 4.2, our model generates denser captions than the ground truth (9.8 vs 7.1 per video). Our strong performance on precision-sensitive metrics (METEOR) and blind pairwise evaluations supports that this reflects valid granular details rather than hallucinations. However, this discrepancy highlights a limitation in current benchmarks, where sparse annotations may undervalue detailed captioning. This creates an inherent evaluation bias that penalizes highly descriptive models. Furthermore, current mainstream benchmarks (*e.g.* YouCook2, ActivityNet) skew heavily towards instructional and sports content. We emphasize the critical need for the development of more comprehensively and densely annotated video benchmarks in the future.

Furthermore, our retrieval mechanism is not restricted to the current corpus. As detailed in the supplementary material, expanding the corpus with instructional texts, *e.g.* HowTo100M (Miech et al., 2019) yields further performance gains (+2.5 CIDEr on ViTT). This suggests that automated, large-scale corpus construction is a promising avenue for enhancing open-world generalization.

Finally, the model may inherit societal biases from its large-scale pretraining datasets, potentially skewing performance across different demographics or activities. Furthermore, given the subjective nature of video captioning and discrepancies in standard evaluation metrics, this model is intended solely for research purposes and not for direct deployment. .

Future Work: While we follow standard community practices by benchmarking on ViTT, YouCook2, and ActivityNet, testing our streaming model on highly complex egocentric datasets, such as COM Kitchens (Maeda et al., 2024), represents a highly relevant future direction for advancing hierarchical clip-level captioning.

## 6 Conclusion

We introduced a novel approach for online dense video captioning that uses a causally-aware autoregressive model to dynamically retrieve and integrate factorized action-object phrases. This aligns the retrieval process with the video's temporal progression, enabling more precise and contextually-grounded captions. Augmented by an effective image-based simulated video pretraining strategy, our method achieves superior caption quality and temporal localization, outperforming state-of-the-art global and online models on the ViTT, YouCook2, and ActivityNet benchmarks.

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

# 7 Appendix

## 7.1 Additional Implementation Details

**CLIP pretraining:** We utilize the CLIP model pretrained on the LAION-2B dataset. Specifically, we use its ViT-Large model (303M parameters) and its 12-layer Transformer text model (128M parameters). The CLIP-initialized ViT is kept frozen throughout all stages of training described below.

**Image captioning pretraining:** We further pretrain the CLIP text model on the image captioning task using the same LAION-2B dataset, with batch size 1024 for 0.2 epochs. We use the Adam optimizer with momentum 0.9, an initial learning rate (LR) of 5e-5, 5000 warmup steps, linear LR decay, weight decay 1e-2.

**Simulated video pretraining:** As described in section 3.4, we apply image-based simulated video pre-training on the entire model, including the newly added Token Aggregator and Autoregressive Transformer modules. To simulate video sequences, we sample 3 to 5 images from the LAION-2B dataset, repeating each image multiple times to form a 16-frame sequence. To create smoother transitions, we blend pixels at the boundaries by applying a weighted sum of two images, using a randomly selected blending ratio $\alpha \in [0.1, 0.9]$, *e.g.*, blending pixels of images A and B as $\alpha A + (1 - \alpha)B$. We apply random augmentations to each frame to avoid overly monotonous sequences. This pretraining follows the same frame-by-frame autoregressive framework for online dense video captioning. We use a batch size of 32 and train the model for 100000 steps. The optimizer is Adam with momentum 0.9, an initial LR of 1e-4, 5000 warmup steps, cosine LR decay, and a weight decay of 1e-5.

**Dense video captioning finetuning:** For dense video captioning, the Token Aggregator and Autoregressive Transformer modules are added. The Token Aggregator is a 4-layer Transformer with 16M parameters, with the last attention pooling layer with $N=32$ queries. The Autoregressive Transformer is a 8-layer Transformer with 32M parameters. In total, the entire model contains approximately 500M parameters. When finetuning on dense video captioning, the model is trained for 20000 steps with a batch size 16. We again use the Adam optimizer with momentum 0.9, an initial LR of 1e-4, 5000 warmup steps, cosine LR decay and a weight decay of 1e-5. We sample $T = 64$ frames per video at $256 \times 256$ resolution. Captions are generated via segment-based decoding every 4 frames (16 total decoding steps). For time tokenization, we use relative time tokens following Vid2Seq (Yang et al., 2023). We quantize a video of duration $T$ frames into equally spaced time bins.

**Mixed training with non-augmented setting:** To facilitate mixed training between augmented and non-augmented settings, we introduce a learnable [none] embedding vector. In the non-augmented setting, this vector replaces the actual text embeddings.

**Inference:** For inference, we follow the Vid2Seq Yang et al. (2023) and Streaming Zhou et al. (2024) to use beam search, with a beam size of 4 and temperature 1.

## 7.2 Statistical Significance

To ensure the reliability of our results, we performed 3 independent runs for our baseline and main models. We report the mean and standard deviation for SODA (S), CIDEr (C), and METEOR (M) across all three

| method | ViTT (S / C / M) | | | YouCook2 (S / C / M) | | | ActivityNet (S / C / M) | | |
|---|---|---|---|---|---|---|---|---|---|
| | S | C | M | S | C | M | S | C | M |
| Baseline (no retrieval) | $9.0_{\pm 0.1}$ | $24.5_{\pm 0.4}$ | $5.6_{\pm 0.1}$ | $6.1_{\pm 0.2}$ | $38.2_{\pm 0.5}$ | $7.1_{\pm 0.2}$ | $6.4_{\pm 0.2}$ | $35.8_{\pm 0.5}$ | $10.1_{\pm 0.2}$ |
| a. Ours | $10.6_{\pm 0.1}$ | $30.3_{\pm 0.3}$ | $7.1_{\pm 0.2}$ | $6.9_{\pm 0.1}$ | $45.6_{\pm 0.4}$ | $7.9_{\pm 0.1}$ | $7.1_{\pm 0.1}$ | $37.6_{\pm 0.3}$ | $11.1_{\pm 0.1}$ |
| b. Ours (a + simulated pretrain) | $11.1_{\pm 0.1}$ | $32.1_{\pm 0.4}$ | $7.6_{\pm 0.1}$ | $7.5_{\pm 0.2}$ | $46.2_{\pm 0.4}$ | $8.2_{\pm 0.1}$ | $7.5_{\pm 0.2}$ | $38.3_{\pm 0.4}$ | $11.9_{\pm 0.2}$ |

Table 12: Statistical significance analysis. We report the mean and standard deviation across 3 independent runs. Ours-a refers to our online action-augmented model, and Ours-b includes simulated video pretraining.

| method | backbone | ViTT | | | | YouCook2 | | | | ActivityNet | | | |
|---|---|---|---|---|---|---|---|---|---|---|---|---|---|
| | | S | C | M | F1 | S | C | M | F1 | S | C | M | F1 |
| a. Ours | CLIP | 10.6 | 30.3 | 7.1 | 39.2 | 6.9 | 45.6 | 7.9 | 33.8 | 7.1 | 37.6 | 11.1 | 54.8 |
| b. Ours (a + simulated pretrain) | CLIP | 11.1 | 32.1 | 7.6 | 40.8 | 7.5 | 46.2 | 8.2 | 34.4 | 7.5 | 38.3 | 11.9 | 55.5 |
| c. Ours (b + tiled frames) | CLIP | 11.4 | 33.5 | 7.9 | 41.1 | 7.9 | 46.8 | 8.5 | 34.6 | 8.0 | 38.9 | 12.8 | 55.8 |

Table 13: Effect of frame contruction strategy.

benchmarks in Table 12. The non-overlapping standard deviation ranges between our models (Ours-a, Ours-b) and the baseline confirm the statistical significance of our reported improvements.

### 7.3 Expanding the Action Phrase Corpus

To improve action-object retrieval, we expand our action phrase corpus by incorporating a broader and more diverse set of action representations. This enhances the richness and generalization of retrieved phrases, leading to improved captioning performance.

**Leveraging instructional text from HowTo-100M dataset.** In addition to action phrases from standard action recognition datasets, we incorporate text subtitles from HowTo100M (Miech et al., 2019), a large-scale instructional video dataset. Rather than using video frames, we focus solely on textual content, extracting concise action phrases that effectively summarize the key activities described.

**Action phrase extraction using an LLM.** To generate structured action phrases, we use Gemma2-27b model (Team et al., 2024) to extract key actions in the format of action-object pairs (*e.g.*, baking ham). The model is prompted to produce succinct descriptions, removing unnecessary details while preserving essential action semantics. An example prompt is: *Your goal is to summarize the input sentence using as few words as possible. Focus on the words describing actions or events. Use singular nouns, avoid articles and numeric terms. Respond in the format of <action verb (ing)> <target object (if any)>. Input: {raw caption}. Answer:*

This caption summarization process filters out irrelevant details, generating a more structured, action-focused corpus. We apply post-processing to refine the corpus by deduplicating similar phrases (*e.g.* minor rewordings or reordered words) and filtering infrequent phrases. This process results in a final corpus of 30,000 action phrases, which we precompute as text embeddings for efficient retrieval. Integrating this expanded corpus improved performance on ViTT over our previous best model, achieving SODA: 11.0 (+0.4), CIDEr: 32.6 (+2.5), METEOR: 7.7 (+0.5), highlighting the effectiveness of the expanded corpus.

### 7.4 Failure Modes.

We observe two primary failure modes in our model. First, under high occlusion, global frame features often fail to capture hidden entities, resulting in missed retrieval of actions or objects and ultimately omitted captions. Second, during fine-grained interactions, the target object's size may be too small relative to the global visual feature, leading the model to predict the correct action but generate a generic or missing object description.

### 7.5 Data Licenses.

The datasets used in this work are under various open licenses suitable for research. LAION: CC-BY 4.0; ViTT: CC-BY-SA; YouCook2: MIT license; ActivityNet-Captions: MIT license.

