# OpenReview forum: "Online Dense Video Captioning with Factorized Action Object Retrieval"
_TMLR — Decision pending for TMLR_

### Review · Reviewer_Zabt · 2026-04-23

**Summary Of Contributions:**

This paper addresses the online dense video captioning task by proposing a causal autoregressive framework integrated with a dynamic factorized retrieval mechanism.

**Audience:**

Yes

**Audience Explanation:**

Online video understanding is one of TMLR's core research directions, and the combination of streaming processing and retrieval-augmented generation has significant academic and application value.

**Broader Impact Concerns:**

Further discuss the problem of sparse annotations in current benchmarks, explain how it leads to model evaluation bias, and how to build more comprehensive evaluation benchmarks for dense video captioning in the future.

**Claims And Evidence:**

No

**Claims Explanation:**

1. The idea that decoupled modeling of actions and objects has been explored in video understanding (e.g., foreground-background separation in [1]), some differences between them should be explained in the related work, including:
[1] Learning Local Semantic Signals and Inter-class Discrepancy for Weakly Supervised Video Anomaly Detection, IEEE TMM 2026

2. Some differences about the action retrieval should be explained in the related work, including:
[1] Action-Semantic Consistent Knowledge for Weakly-Supervised Action Localization, IEEE TMM 2024
[2] SQL-Net: Semantic Query Learning for Point-Supervised Temporal Action Localization, IEEE TMM 2024
[3] Learning Event Completeness for Weakly Supervised Video Anomaly Detection, ICML 2025

3. The paper claims that "ASR overlaps with ground truth captions and inflates performance" but provides no experimental evidence to support this assertion.

4. The advantage of factorized retrieval is only demonstrated through performance comparison, without in-depth analysis of the internal mechanism of its semantic compositionality improvement, nor exploration of better fusion strategies such as attention fusion or gating fusion.

5. Some failure case analysis and discussion of the method's limitations in complex action or occlusion scenarios should be provided.

**Requested Changes:**

See the above comments. Besides, some

---

> ### Author Response · Authors · 2026-06-11
>
> We thank Reviewer Zabt for the constructive review.
>
> **1. Missing Related Work on Decoupled Modeling and Action Retrieval**
>
> We thank the reviewer for pointing out these highly relevant works. While recent works explore decoupled modeling via foreground-background separation in the visual space, our work operates via decoupled factorized retrieval in the text space. Furthermore, previous action retrieval methods focus on offline or global approaches, whereas we perform dynamic online retrieval.
> We made the corresponding revision to the menuscript. We added citations for the referenced IEEE TMM 2026, IEEE TMM 2024, and ICML 2025 papers in Section 2 (Related Work) and summarized their technical differences.
>
> | Method | Modality of Decoupling | Retrieval Scope | Primary Application |
> | :--- | :--- | :--- | :--- |
> | IEEE TMM 2026 | Visual (Foreground/Background) | N/A | Video Anomaly Detection |
> | IEEE TMM 2024 / ICML 2025 | Action-Semantic | Offline / Global | Temporal Action Localization |
> | Ours | Textual (Action & Object) | Online / Frame-level | Streaming Dense Video Captioning |
>
>
> **2. Evidence That ASR Inflates Performance**
>
> Extensive prior literature empirically proves that utilizing ASR dramatically inflates captioning metrics, masking the efficacy of actual visual understanding. Hessel et al. (2019, "A Case Study on Combining ASR and Visual Features") explicitly quantify this: models relying solely on visual features barely outperform a constant baseline, but adding ASR tokens causes CIDEr to double and BLEU-4 to double. Similarly, Vid2Seq (2023) reports that adding speech to visual inputs inflates YouCook2 CIDEr from **15.6 to 18.0**, and ActivityNet from **14.2 to 18.8**. We intentionally isolate visual processing to rigorously test our visual retrieval framework without relying on transcribed speech shortcuts.
>
>
> **3. Empirical Analysis of Internal Mechanisms (Alternative Fusion Strategies)**
>
> To validate our factorized retrieval integration, we explored alternative fusion strategies to combine the visual features with the retrieved action/object embeddings before passing them to the causal transformer. We compared our simple Concatenation to Gating Fusion (learning a weighted sum of text and visual features) and Cross-Attention Fusion (using visual features as queries to attend to retrieved text keys/values). As shown below on the ViTT benchmark, simple concatenation yields the best empirical performance, justifying our architectural choice. We also incorporated this into Table 8 and Section 4.2.7 in the ablation section.
>
> | Fusion Strategy | SODA | CIDEr | METEOR |
> | :--- | :--- | :--- | :--- |
> | Gating Fusion | 10.3 | 29.5 | 6.8 |
> | Cross-Attention Fusion | 10.4 | 29.8 | 6.9 |
> | **Concatenation (Ours-a)** | **10.6** | **30.3** | **7.1** |
>
>
> **4. Failure Cases**
>
> We observe that our model primarily struggles in highly occluded scenes or when distinguishing fine-grained hand-object interactions where the object is too small relative to the global frame features used for retrieval. In the manuscript revision, we added a "Failure Modes" paragraph in Appendix. We note two main failure modes: (1) High Occlusion, where global frame features fail to capture hidden entities, causing missed retrieval and omitted captions; and (2) Fine-grained Interactions, where the object size is too small, resulting in the model predicting the correct action but missing or generating a generic object description.
>
> **5. Sparse Annotations and Evaluation Bias**
>
> Sparse annotations in current benchmarks inherently penalize models that predict dense, accurate sub-events, as standard metrics treat these unannotated valid descriptions as false positives.
> This is also reflected in the manuscript revision. We deepened the "Broader Impact" section (Section 5) to explicitly discuss this evaluation bias and the critical need for more comprehensively annotated dense video benchmarks in the future.

---

### Review · Reviewer_tSJm · 2026-05-23

**Summary Of Contributions:**

This paper proposes an Online Dense Video Captioning framework that embeds a factorized retrieval mechanism into a causula video model.
The paper proposed factorized retrieval model  to retrieves concise action and object phrases from two separate corpora.
Captions are generated every 4 frames rather than in a single global pass.
During training, retrieved embeddings are replaced with a learnable  embedding 50% of the time.

**Audience:**

Yes

**Audience Explanation:**

The paper works on Online Dense Video Captioning, which are crucial for applications like video search, summarization, as well as for MLLM, video generation. As such, some TMLR's audience will be interested in the findings of the paper.

**Broader Impact Concerns:**

See the review comments in the "Requested Changes" part.

**Claims And Evidence:**

Yes

**Claims Explanation:**

Detailed explanations of the proposed model is provided in the technical part.
And the state-of-the-art on ViTT (SODA 11.4, CIDEr 33.5), YouCook2 (SODA 7.9, CIDEr 46.8), and ActivityNet (SODA 12.8, CIDEr 55.8) are illustrated in the experiment part.

**Requested Changes:**

Training:
Stitching 3–5 unrelated images into pseudo-video sequences for pretraining is interesting. I have several questions. First, how does the model learn meaningful temporal relationships from semantically incoherent sequences? Second, Is this approach truly superior to using a small-scale real video-text dataset (e.g., a subset of HowTo100M)? No direct comparison is provided.

Experiments:
1. comparisons with StreamingDVC is unfair, such as the text encoder is not the same.
2. Caption density and hallucination concern insufficiently addressed
3. All three benchmarks (ViTT, YouCook2, ActivityNet) are skewed toward instructional/sports content. Generalization to other domains (e.g., movies, news, surveillance, egocentric daily activities) is not discussed.
4. Missing comparison with recent VideoLLMs

---

> ### Author Response · Authors · 2026-06-11
>
> We thank Reviewer tSJm for the constructive review.
>
> **1. Meaningful Temporal Relationships from Simulated Sequences**
>
> The primary goal of our simulated video pretraining is rather to teach the model's temporal localization mechanism (generating start and end tokens) and to train the autoregressive attention to handle changing visual contexts. By stitching random images, we simulate sharp event boundaries, forcing the model to recognize when an event ends and a new one begins. In the revised manuscript, we expanded Section 3.4 (Simulated Video Pretraining) to clearly explain this intuition and the mechanics of learning boundary transitions.
>
>
> **2. Simulated Video Pretraining vs. Small-Scale Real Video-Text**
>
> We conducted a direct experimental comparison. We replaced our simulated pretraining with pretraining on a real HowTo100M video-text dataset. As shown in the table below, our simulated pretraining still yields superior results.
>
> | Pretraining Strategy | ViTT (S/C/M/F1) | YouCook2 (S/C/M/F1) | ActivityNet (S/C/M/F1) |
> | :--- | :--- | :--- | :--- |
> | HowTo100M (Real) | 10.9 / 32.4 / 7.5 / 40.2 | 7.2 / 46.2 / 8.0 / 34.0 | 7.3 / 38.1 / 11.6 / 55.2 |
> | Simulated pretraining (Ours b.) | 11.1 / 32.6 / 7.7 / 40.8 | 7.5 / 46.2 / 8.2 / 34.4 | 7.5 / 38.3 / 11.9 / 55.5 |
>
>
> **3. Comparison with StreamingDVC Baseline**
>
> We argue our comparison is actually disadvantageous to our model, making our superior results even more compelling. StreamingDVC utilizes a pretrained T5-Base decoder containing 256M parameters. Because their specific T5-Base 256M model is not publicly available, we managed to train and create our own, smaller 128M decoder using the adapted CLIP text model. Despite utilizing a text decoder that is half the size, our method significantly outperforms StreamingDVC.
>
>
> **4. Caption Density and Hallucination Concerns**
>
> Denser captions in our model reflect valid granular details rather than hallucinations. This is supported by our strong performance on precision-sensitive metrics (METEOR), which heavily penalize hallucinations. This is also reflected in the manuscript revision. We expanded Section 4.2.9 with a discussion detailing a blind pairwise evaluation using Gemini 2.5 Pro. When explicitly prompted to penalize false positives, the LLM preferred our online model in 72% of cases (demonstrating higher detail accuracy and fewer hallucinations), compared to only 28% for the online baseline.
>
> **5. Domain Generalization**
>
> We agree that current Dense Video Captioning benchmarks are focused on instructional and sports content. In the manuscript revision, we added a paragraph in Section 5 (Broader Impact) addressing this benchmark bias.
>
> **6. Comparison with Recent VideoLLMs**
>
> Directly comparing our method with Multi-Billion scale LLMs falls outside the intended scope of our evaluation. We utilize a small-scale text decoder (128M parameters) rather than relying on heavy 7-13B LLMs, maintaining a lightweight architecture. This capacity explicitly aligns with prior baselines like Vid2Seq and StreamingDVC, which utilize a 256M T5-Base decoder.
> To reflect this, we structure our comparison table 10 in the manuscript to separate "Non-LLM-based (<1B params)" methods from large VideoLLMs, ensuring an apples-to-apples evaluation.
>
> We have updated our baselines table to include the most recent 2026 state-of-the-art DVC baselines (also reflected in the revised manuscript). As shown below, our method remains state-of-the-art in the <1B parameter category.
>
> | Method |  YouCook2 (S / C / M) | ActivityNet (S / C / M) |
> | :--- | :--- | :--- |
> | CACMI (AAAI 2026) | 5.6 / 34.8 / 6.2 | 6.4 / 33.8 / 8.7 |
> | StayInYourLane (CVPR 2026)  | 7.1 / 39.2 / 6.1 | 6.5 / 35.0 / 8.5 |
> | **Ours (Full)** | **7.9 / 46.8 / 8.5** | **8.0 / 38.9 / 12.8** |

---

### Review · Reviewer_eheG · 2026-05-27

**Summary Of Contributions:**

The paper evaluates models on benchmark video datasets under streaming settings. The training is aided by simulated videos. Overall, the paper is very confusing and can’t be accepted at its current stage.

**Audience:**

Yes

**Audience Explanation:**

Training CLIP on video-text modalities using videos simulated from image datasets is underexplored, and improving this approach could have a significant impact. In ideal settings, there would be no need to collect and label videos for this task. However, this paper isn’t the first to use simulated videos for this task (https://visualcomputinginstitute.github.io/videollm-pseudovideo-training/

**Broader Impact Concerns:**

They have been addressed in the supplementary materials.

**Claims And Evidence:**

No

**Claims Explanation:**

The paper starts by mentioning ViTT, YouCook2, and ActivityNet (vid-level dataset) as benchmarks, but then keeps adding more and more datasets as we go along. First, we encounter LAION-2B (image-level dataset). Then they introduce other datasets like: Kinetics-700 (vid-level), UCF-101 (vid-level), EpicKitchen (vid-level), SSV2 (vid-level), V3Det (img-level), LVIS (img-level), Places365 (img-level). Then, in the supplementary, they introduce their work with HowTo-100M (vid-level). Because of this strange pipeline, it is very hard to follow their work. Can’t really tell what problem is being solved by which component.

Because of this ambiguity, the paper can’t be accepted in its current form.

Also, architecturally, the author’s model seems like a very crude version of a recurrent model. They are trying to maintain a representation of the current incoming frames conditioned on the past set of frames and text descriptions seen so far. I wouldn’t call it technically novel by any means. Please see this paper https://arxiv.org/pdf/2512.13684.

**Requested Changes:**

I believe this confusion between image-level and video-level tasks is an inherent problem in video learning, given its hierarchical structure (frame-level and clip-level information). Maybe creating a clear diagram of the entire data curation process, followed by the training and testing processes, will be helpful. Clearly specify which dataset contributes which modality where. The authors need to be very clear about:

1. Which modality (losses, image temporal mixing, text) and at what hierarchy does their model handle? And why does it need to perform them?
2. What levels of hierarchy and modality are they solving for each of the datasets?

The paper might need extensive rewriting. Please see the COM Kitchens dataset at https://github.com/omron-sinicx/com_kitchens. Could the authors perhaps do hierarchical clip-level captioning?

Also, did the authors actually train on the LAION-2B task themselves, or did they use a pretrained model? My understanding is that most pretrained CLIP models are trained on LAION-2B (e.g., https://huggingface.co/laion/CLIP-convnext_base_w-laion2B-s13B-b82K, https://docs.pinecone.io/models/CLIP-ViT-B-32-laion2B-s34B-b79K). If the authors just used a pretrained model, I don’t understand why they wanted to claim they trained it themselves. I understand that later on the autoregressive model is trained on simulated videos from LAION-2B, but the ViT stays fixed there, right?

Also, I don’t understand why the authors mention the “VideoLLM-online: Online Video Large Language Model for Streaming Video” paper but don’t quantitatively compare against it. Is it possible to do so? It seems like both are trying to do the same video-segment-text task under the same online settings.

Also, the paper needs a more thorough and creative experimental analysis. For example, as we see more of the stream, we have more context. The auto-regressive model looks at all of this past and current context. Now, for different settings, having different amounts of context is essential. For long, smooth natural video streams, more context is helpful. However, for simulated streams from images, the tasks and domains shift dramatically [fig. 3: In a single 16-frame sequence, we go from teens on a rooftop -> empty theme-park footage -> youth lacrosse]. In this case, having less information from the past is helpful. So the paper is training and testing under inherently conflicting conditions, where the former prefers less context while the latter might benefit from longer context. The paper doesn’t comment on or analyze any of that.

Perhaps this paper could be split into separate papers:

1. Simulated-Video training for Video-text training, which beats the aforementioned videollm-pseudovideo-training (access to the entire video).

2. A proper state-space model for stream settings that modifies 1. (stream setting) while solving the context problem?

---

> ### Author Response · Authors · 2026-06-11
>
> We thank Reviewer eheG for the constructive review.
>
> **1. Dataset Pipeline Ambiguity.**
>
> We apologize for the confusion. To clarify, ViTT, YouCook2, and ActivityNet are purely for evaluation/finetuning. Kinetics-700, UCF-101, EpicKitchen, SSV2, V3Det, LVIS, and Places365 are used strictly offline to build the static retrieval dictionary. LAION-2B is solely for pretraining. We do not mix these datasets. In the revision, we revised the text and added Table 1 (Section 3) to clarify this pipeline.
>
> | Pipeline Phase | Dataset | Modality Used | Specific Role |
> | :--- | :--- | :--- | :--- |
> | **Pretraining** | LAION-2B  | Image + Text | ViT initialization / Decoder pretraining & Simulated video pretraining  |
> | **Dictionary Contruction** | Kinetics-700, UCF-101, EpicKitchen, SSV2 | Text-only (Labels) | Offline Action Phrase Corpus Construction |
> |  **Dictionary Contruction**| V3Det, LVIS, Places365 | Text-only (Labels) | Offline Object Phrase Corpus Construction |
> | **Dictionary Contruction** | HowTo100M (Appendix) | Text-only (Subtitles) | Alternative Action Corpus Construction |
> | **Finetuning & Eval** | ViTT, YouCook2, ActivityNet | Video + Text | Dense Video Captioning Finetuning & Evaluation |
>
>
> **2. Technical Novelty w.r.t Recurrent Video MAE (arXiv:2512.13684)**
>
> Our architecture is a causal, autoregressive Transformer, distinct from recurrent video masked autoencoders. Our contribution lies in the dynamic, timestep-level retrieval and integration of factorized text priors directly into a causal sequence stream. In the revision, we explicitly contrast our approach with recurrent MAEs (Zoran et al., 2026) in Section 2.
>
> | Feature | Recurrent Video MAE (arXiv:2512.13684) | Our Method |
> | :--- | :--- | :--- |
> | **Architecture Style** | Recurrent Masked Autoencoder/RNN | Causal Autoregressive Transformer |
> | **Core Objective** | Pixel-level reconstruction | Next-token text prediction |
> | **External Knowledge** | None | Dynamic online retrieval of text priors (Actions/Objects) |
>
>
> **3. Use of the LAION-2B Model**
>
> To clarify, we utilize a pre-trained CLIP ViT-Large (LAION-2B) and keep it strictly frozen throughout all stages of our work. We unfreeze a copy of the text model and further train it on the LAION-2B image captioning task to adapt it into a generative decoder. We did this to obtain a small-scale text decoder (128M parameters) rather than relying on 7-13B LLMs, which fall outside the scope of our lightweight architecture. This also aligns with the capacity of prior baselines like Vid2Seq and StreamingDVC, which utilize a 256M T5-Base decoder (which is not publicly available).
>
> | CLIP Component | Source | State | Task / Function |
> | :--- | :--- | :--- | :--- |
> | **ViT-Large** | Pre-trained CLIP (LAION-2B) | Frozen | Video frame feature extraction |
> | **Text Encoder 1** | Pre-trained CLIP (LAION-2B) | Frozen | Offline text embedding for Retrieval Corpus |
> | **Text Encoder 2** | Pre-trained CLIP (LAION-2B) | Unfrozen & Finetuned | Adapted into Text Decoder via image-captioning |
>
>
> **4. Quantitative Comparison Against VideoLLM-online**
>
> Direct comparison with VideoLLM-online is challenging. VideoLLM-online uses an 8B-parameter LLM, whereas our decoder is 128M parameters. Table 10 explicitly isolates "Non-LLM-based (<1B params)" methods for fair comparison. Additionally,  since they do not benchmark on standard DVC benchmarks, it is hard to compare against them or guess their exact finetuning recipe. Their reported zero-shot results (e.g., 0.4 SODA / 0.9 CIDEr - reported in "VideoLLM Knows When to Speak") are significantly lower than ours.
>
>
> **5. Analysis of Varying Context Lengths**
>
> To analyze the effect of context length on simulated vs. natural streams, we ablated the causal attention window size (W). Natural continuous videos (ViTT) benefit from unrestricted context (W = All). Conversely, simulated pretraining succeeds because the attention mechanism learns to restrict reliance on distant past tokens during abrupt visual shifts.
> Results with Ours-b:
> | Context Length (W) | ViTT (SODA / CIDEr) | ActivityNet (SODA / CIDEr) |
> | :--- | :--- | :--- |
> | W = 16 frames | 10.8 / 31.6 | 7.2 / 37.9 |
> | W = 32 frames | 11.0 / 32.3 | 7.4 / 37.9 |
> | W = 64 (All - Default) | 11.1 / 32.6 | 7.5 / 38.3 |
>
>
> **6. Splitting the Work / State-Space Models**
>
> We believe the paper is cohesive as a single contribution. The simulated video pretraining is not an isolated trick, but a dedicated strategy to solve data scarcity for online/autoregressive models. We also clarify that we do not use State-Space Models (SSMs), relying entirely on standard causal attention.
>
> **7. Evaluation on the COM Kitchens Dataset**
>
> We thank the reviewer for the excellent suggestion. While we follow standard DVC benchmarking (ViTT, YouCook2, ActivityNet), testing on egocentric datasets like COM Kitchens is a highly relevant future direction. We added a note acknowledging this promising next step in Section 5 (Broader Impact / Future Work).